

# Rainfall-induced shallow landslides and soil wetness: comparison of physically-based and probabilistic predictions

Elena Leonarduzzi[1,2], Brian W. McArdell[2], and Peter Molnar[1]

[1]Institute of Environmental Engineering, ETH Zurich, Zurich, Switzerland
[2]Swiss Federal Institute for Forest, Snow and Landscape Research WSL, Birmensdorf, Switzerland

**Correspondence:** Elena Leonarduzzi (leonarduzzi@ifu.baug.ethz.ch)

**Abstract.** Landslides are an impacting natural hazard in alpine regions, calling for effective forecasting and warning systems. Here we compare two methods (physically-based and probabilistic) for the prediction of shallow rainfall-induced landslides in an application to Switzerland, with a specific focus on the value of antecedent soil wetness. First, we show that landslide susceptibility predicted by the factor of safety in the infinite slope model is strongly dependent on soil data inputs, limiting the hydrologically active range where landslides can occur to only ∼20% of the area with typical soil parameters and soil depth models. Second, the physically-based approach with a coarse resolution model setup (TerrSysMP) 12.5km×12.5km downscaled to 25m×25m with the Topographic Wetness Index to provide water table simulations for the infinite slope stability model did not succeed in predicting local scale landsliding satisfactorily, despite spatial downscaling. We argue that this is due to inadequacies of the infinite slope model, soil parameter uncertainty, and the coarse resolution of the hydrological model. Third, soil saturation estimates provided by a higher resolution 500m×500m conceptual hydrological model (PREVAH) provided added value to rainfall threshold curves for landslide prediction in the probabilistic approach, with potential to reduce false alarms and misses. We conclude that although combined physically-based hydrological-geotechnical modelling is the desired goal, we still need to overcome problems of model resolution, parameter constraints, and landslide validation for successful prediction at regional scales.

## 1 Introduction

Landslides are a natural hazard affecting alpine regions worldwide. They damage infrastructure, buildings, sometimes leading to loss of life (e.g., Kjekstad and Highland, 2009; Salvati et al., 2010; Petley, 2012; Trezzini et al., 2013; Mirus et al., 2020). Shallow landslides occur when and where the applied shear on the soil-bedrock interface exceeds the shear strength of the soil on a slope. Their occurrence is determined by two key factors: predisposing factors, which are a collection of soil and land surface properties of a certain location which make it susceptible or not to landsliding (e.g., Reichenbach et al., 2018); and triggering factors, which are those that initiate slope failure on susceptible slopes. In general, most landslides are either triggered by earthquakes or rainfall (Iverson, 2000; Highland et al., 2008; Leonarduzzi et al., 2017; Marc et al., 2019, e.g.,). Here we focus on shallow rainfall-induced landslides, which involve the top layer of the soil, typically less than 2m thick, and fail instantaneously. In such landslides, failure is typically the result of the development of positive pore water pressure in the





soil, which decreases its strength (e.g., Anderson and Sitar, 1995; Highland et al., 2008). This condition is often associated with intense or long lasting rainfall events that saturate the soil by vertical infiltration and lateral subsurface drainage. The wetness of the soil prior to the triggering rainfall is therefore also important (Bogaard and Greco, 2018).

Several approaches exist for the prediction of landslides that focus on one or more predisposing and triggering factors, typically classified into 3 types: susceptibility mapping, probabilistic approaches, and physically-based modelling (e.g., Aleotti 30  and Chowdhury, 1999).

Susceptibility mapping assesses the susceptibility of a certain area to landsliding based on predisposing factors. In statistical susceptibility mapping, the different predisposing factors, geological, topographical, and climatological properties, are combined and used as explanatory variables in a statistical model (e.g., Reichenbach et al., 2018). Landslide hazard maps are then generated by various forms of linear and nonlinear multivariate regression models (e.g., Chung et al., 1995), logistic regression 35  (e.g., Ohlmacher and Davis, 2003; Ayalew and Yamagishi, 2005; Lee and Pradhan, 2007; Yilmaz, 2009; von Ruette et al., 2011), or machine learning algorithms (e.g., Saito et al., 2009; Ermini et al., 2005; Yilmaz, 2009). Susceptibility mapping can also be achieved by applying a physically-based geotechnical model which identifies the likelihood of failure in a region based on an assessment of likely soil water distribution in space (e.g., Baum et al., 2002, 2008; Dietrich and Montgomery, 1998; Formetta et al., 2016).

Probabilistic approaches focus mainly on the temporal component of the landslide hazard (triggering factors), rather than the spatial susceptibility (predisposing factors). They are based on the assumption that rainfall is the main triggering factor, and take advantage of historical records of rainfall and landslides. These databases are combined to learn which meteorological conditions have been associated with the triggering of landslides in the past. This allows then to recognise critical conditions in weather forecasts of the coming days and estimate how likely the occurrence of landsliding is. The most common of these 45  approaches is that of rainfall thresholds, and in particular intensity-duration or total rainfall-duration threshold curves (e.g., Guzzetti et al., 2007; Leonarduzzi et al., 2017; Segoni et al., 2018). While rainfall is mostly the main triggering factor, soil wetness conditions prior to triggering rainfall can also be included in this framework (e.g., Bogaard and Greco, 2018; Marino et al., 2020). The antecedent soil wetness conditions can be derived in many different ways, each with its advantages and limitations, for example from in-situ measurements (depend on network density, e.g., Wicki et al., 2020), remote sensing of 50  soil moisture (suffer from low resolution and insufficient penetrating depth, e.g., Brocca et al., 2012; Thomas et al., 2019), through proxies of soil wetness like antecedent rainfall (miss evapotranspiration and snowmelt, e.g., Glade et al., 2000; Godt et al., 2006; Mathew et al., 2014), or by hydrological soil water balance modelling (e.g., Ponziani et al., 2012; Thomas et al., 2018).

Finally, physically-based modelling approaches are usually made up of two components to simulate slope stability in time 55  and space: an hydrological and a geotechnical model. The hydrological model is used to estimate the condition of the soil, i.e. the pore water pressure and/or saturation, which are then used in the geotechnical model for the estimation of slope stability (e.g., by the infinite slope or other hydromechanical slope failure model). These approaches are theoretically the most accurate and predict both when and where a landslide could occur, but are typically very computationally expensive and data demanding.





For these reasons, they are typically applied on individual slopes in landslides-prone areas or small catchments (Cohen et al.,
2009; von Ruette et al., 2013; Anagnostopoulos et al., 2015; Fan et al., 2015, 2016).

In this work, we conduct a comparison of a probabilistic and physically-based modelling approach to landslide prediction
with the specific question of the value of the inclusion of antecedent soil wetness state in the prediction. Our scale of analysis
is regional (Switzerland) instead of hillslope/catchment scale, because it is at this scale that landslide early warning systems
need to be developed (e.g., Staehli et al., 2015). First we explore the regional susceptibility to landslides following the infinite
slope approach (physically-based susceptibility mapping). This allows us to understand where hydrology can play a role in the
landscape in triggering landslides, i.e. identifying areas where the transient soil wetness results in the Factor of Safety (FoS)
fluctuating above 1 (stable) and below 1 (unstable). We then explore two approaches to account for the soil wetness state for
landslide prediction, taking advantage of the hydrological estimates of soil moisture provided by two different models set-up
for forecasting purposes and covering Switzerland.

(1) A fully physically-based approach that takes advantage of a state-of-the-art European simulation of hydrology (Furusho-
Percot et al., 2019) with three physically-based coupled models (climate forecast model, land surface model, hydrological
model) at a coarse resolution (12.5km×12.5km), from which we extract water table depth at daily resolutions. We downscale
these simulations to a higher spatial resolution more appropriate for landslide forecasting following the Topographic Wetness
Index method (e.g., Blöschl et al., 2009; Beven, 1995). The high resolution groundwater field is then used as a dynamic
component in the Factor of Safety estimation in the infinite slope approach. This framework is designed based on similar
existing blueprints for landslide warning systems (Schmidt et al., 2008; Wang et al., 2020).

(2) A probabilistic approach in which we develop rainfall threshold curves for landslide prediction based on a combination
of historical databases of rainfall and landslides for Switzerland (Leonarduzzi et al., 2017). We then combine these predictions
with estimated soil saturation by a Swiss operational, semi-distributed conceptual hydrological model (PREVAH,  Viviroli
et al., 2009) to quantify whether there is a strong signal in antecedent soil moisture which could be used in rainfall threshold
curve methods for landslide prediction at this scale.

The comparison between the two approaches allows us to answer the following questions: 1) Is the infinite slope approach
valuable for landslides hazard assessment at the regional scale? 2) Where does hydrology play a role in the triggering of
landslides in Switzerland? 3) Can we generate a computationally feasible physically-based setup for landslide forecasting and
early warning at the regional scale? 4) Does soil wetness help improve prediction of landslides based on rainfall thresholds? 5)
Which approach is preferable for landslide prediction and why?





## 2 Methods and Data

### 2.1 Physically-based approach

#### 2.1.1 The infinite slope model

For the stability assessment, we choose to follow the infinite slope approach, because this is one of the most widely used models for slope failure prediction (e.g., Pack et al., 1998; Iverson, 2000; Baum et al., 2008; Lu and Godt, 2008). It is based on the assumption that the thickness of the sliding mass (soil) is much smaller than the length of the slope, which is typically true for shallow landslides (up to 2m deep). The Factor of Safety (FoS) is computed as the ratio between soil shear strength and applied stress to the soil layer:

$$FoS(t) = \frac{c + [\gamma d - \gamma_w h] cos^2 \beta \tan\phi}{\gamma d sin\beta cos\beta} \qquad (1)$$

where $h$ is the water pressure head within the soil layer [m] (see Section 2.1.3) $d$ the soil depth [m] (see Section 2.1.2), $c$ is cohesion [Pa], $\gamma$ is soil unit weight [N/m$^2$] (computed from the bulk soil density $\rho$ and gravitational acceleration $g$ as $\gamma = \rho * g$) , $\gamma_w$ is the specific weight of water [N/m$^2$], $\beta$ is the slope angle [rad], and $\phi$ is the internal friction angle [rad]. Typically FoS=1 is assumed to be the threshold failure value, with landsliding occurring when FoS<1, i.e. when the applied shear stress exceeds

the soil shear strength.

All calculations are done at the resolution of the DEM, that is a grid of cell size 25m×25m. This resolution is a result of testing (not reported here), and a compromise between not violating the infinite slope assumptions (length scale of landslides >> their depth), keeping the grid size similar to that of a typical landslide detachment area, but also capturing local topographic gradients $\beta$, which are smoothed as resolution decreases.

To estimate the bulk soil density, cohesion, and friction angle we use publicly available datasets OpenLandMap (OpenLandMap; Hengl et al., 2017), which provide global maps of a wide range of soil, land cover, hydrology, geology, climatic and relief characteristics. All the maps used here are available at a resolution of ca. 250m×250m. The soil properties are provided for 6 different depths up to 200 cm produced by machine learning algorithms trained on soil profiles globally (SoilGrids Dataset). For the estimation of the bulk soil density, we compute the thickness-weighted average for the 2m soil column. For the

friction angle, we first associate a value for each soil texture class (USDA system) present in Switzerland in the OpenLandMap dataset (Geotechdata.info, Angle of Friction). Then at each location, we choose the value of the friction angle at the depth corresponding to the local soil depth (e.g. if the soil is 120cm deep locally, the closest value in SoilGrids will be that at 1m depth). Finally for cohesion, we assume that the soil itself is cohesionless ($c = 0$), but we add the important contribution of vegetation to cohesion on slopes. From the landcover map in OpenLandMap, we identify 8 classes of tree cover, and assign them

a cohesion value between $c = 5 - 22$ kPa. Denser tree covers and mixed forests are associated with larger values of cohesion (Schwarz et al., 2012; Dorren and Schwarz, 2016). To quantify the sensitivity of the FoS estimation to the vegetation-driven cohesion, we also simulate the reference case in which cohesion is assumed $c = 0$ over the entire country (Figure 1).



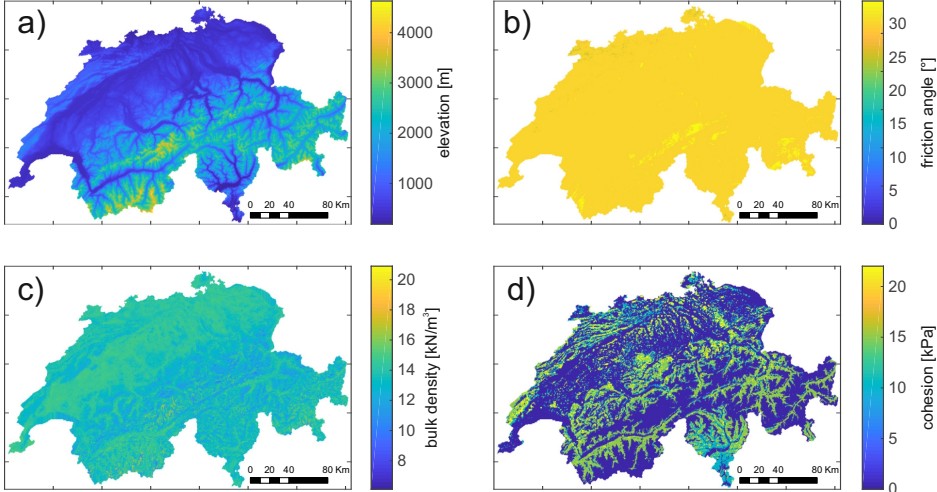

**Figure 1.** Maps of the distributed input used in the Factor of Safety calculations. a) The 25m digital elevation model (Swisstopo), b) friction angle obtained from the OpenLandMap USDA texture class and provided soil depth, c) bulk density obtained from OpenLandMap, and d) cohesion estimated for the land cover map from OpenLandMap. The friction angle depends on the local soil depth, here the soil depth estimated with the linear diffusion model is shown.

To assess the susceptibility to landsliding in Switzerland, we compute the Factor of Safety of the two end-member scenarios: completely wet soil ($h = d$) or completely dry soil ($h = 0$), which give us the minimum and maximum FoS. This allows us to
identify unconditionally stable and unstable areas in our domain just based on hydrology. Unconditionally stable areas have the minimum FoS>1 for a completely wet soil and will never fail regardless of the actual hydrological state. Unconditionally unstable areas have the maximum FoS<1 for a completely dry soil and will (should) always fail according to Eq. 1. In all other areas, hydrology can play a role in the initiation of landslides according to the FoS methodology.

We then compute the dynamic Factor of Safety in time and its statistics for all cells (25m×25m) in which at least one
landslide was recorded according to the landslide database (for details on the database see Section 2.2), using the water pressure head $h$ estimated from the hydrological model described in Section 2.1.3. Additionally, we also compute the average FoS during each triggering and non-triggering rainfall event which are defined in Section 2.2. These analyses allow us to observe variations in FoS at cell level and relate them to observed landslide occurrence. If these relations are found to be strong, we hypothesize that a warning system could be set up based on the estimated FoS, combined with forecasted rainfall.





### 2.1.2 Soil depth

Because soil depth is the most poorly known variable and uncertain parameter in the slope stability model in Eq. 1, here we use four different methods to estimate soil thickness distributions in space and test their impacts on FoS estimates: (1) Uniform soil depth of 1 m for the entire country. (2) Slope-dependent model (Saulnier et al., 1997):

$$d_i = d_{max} \left\{ 1 - \left[ \frac{tan\beta_i - tan\beta_{min}}{tan\beta_{max} - tan\beta_{min}} \left( 1 - \frac{d_{min}}{d_{max}} \right) \right] \right\} \tag{2}$$

where $d_{max}$ is maximum soil depth, $d_{min}$ the minimum soil depth (assumed to be 5cm), $\beta_i$ is the local slope, $\beta_{max}$ is the maximum slope above which no soil layer can form (assumed to be $45°$), and $\beta_{min}$ the minimum slope ($0°$). (3) Elevation-dependent model (Saulnier et al., 1997):

$$d_i = d_{max} - \frac{z_i - z_{min}}{z_{min} - z_{max}(d_{max} - d_{min})} \tag{3}$$

where $z_i$ is the local elevation, $z_{max}$ is the maximum elevation, and $z_{min}$ the minimum elevation. (4) Steady state soil depth produced by the linear diffusion transport model (Roering, 2008) where we simulate the distributed soil depth after 15'000 years of soil development. This approach is based on mass conservation (Eq. 4), with soil production decreasing exponentially with soil depth (Eq. 5) and soil erosion and transport assumed to be linearly dependent on slope (Eq. 6).

$$\frac{\partial d_i}{\partial t} = -\nabla \boldsymbol{q_{s,i}} + \frac{\rho_r}{\rho_s} \epsilon_i \tag{4}$$

$$\epsilon_i = \frac{\epsilon_0}{cos\beta_i} e^{-\mu d_i cos\beta_i} \tag{5}$$

$$\boldsymbol{q_{s,i}} = -K_l \nabla z_i \tag{6}$$

where $\frac{\partial d_i}{\partial t}$ is the change of soil depth in time, $\boldsymbol{q_{s,i}}$ the transport vector at location i, $\frac{\rho_r}{\rho_s}$ the ratio between the bedrock and soil density (2 as in Dietrich et al., 1995), $\epsilon_i$ the soil production rate at location i, $\epsilon_0$ the maximum soil production rate associated with 0 depth (0.000268 m/year as in Heimsath et al., 2001), $\beta$ the slope at location i, $\mu$ the critical value depth (3 1/m as in Roering, 2008), $K_l$ the coefficient of linear proportionality (0.0050 as in Dietrich et al., 1995), and $\nabla z_i$ the gradient of elevation at location i.

For the soil depth models (2)-(4), to be consistent, we fix the maximum soil depth $d_{max} = 2$m.





### 2.1.3 Hydrology

The water pressure head within the soil layer required for the calculation of the Factor of Safety ($h$ in Eq. 1), is provided by an operational European forecasting system. This consists of the climatology from 1989 to current day, obtained by applying the

Terrestrial Systems Modeling Platform (TerrSysMP) (Kurtz et al., 2016). This platform is made up of three physically-based coupled models that solve the water and energy fluxes from the atmosphere to the groundwater: a weather prediction model, a land surface process model, and a hydrological model for surface and subsurface 3D water fluxes. TerrSysMP is produced at daily resolution over a 12.5km×12.5km grid covering Europe. Several state and flux variables are available and can be freely accessed, here we use the water table depth from the surface ($wtd$, positive when the water table is above the surface).

The coarse 12.5km×12.5 km simulations of $wtd$ by TerrSysMP have to be downscaled to the 25m×25m grid of the geotechnical model so that the high resolution topography which affects the lateral redistribution of soil water (groundwater) is accounted for. We do this using the Topographic Wetness Index (TWI) approach, following the blueprints of previous studies (Schmidt et al., 2008; Wang et al., 2020). This appraoch assumes that the local departure of the water table to the coarser scale average is proportional to the departure of the local TWI from the spatial average (Eq. 7 and Eq. 8):

$$wtd_i(t) = \overline{wtd}_j(t) + \frac{1}{f}\left(TWI_i - \overline{TWI}_j\right) \tag{7}$$

$$TWI_i = \ln\left(\frac{a_i}{\tan\beta_i}\right) \tag{8}$$

where the index $i$ refers to the local 25m×25m cell considered, and $j$ to the 12.5km×12.5km cell from TerrSysMP in which cell i is embedded, $wtd_i(t)$ is the $TWI$-redistributed local water table depth at time $t$, $\overline{wtd}_j(t)$ is the coarser scale water table depth estimated by TerrSysMP in $j$ at time $t$, $f$ is a scaling parameter (here assumed to be 1), $TWI_i$ is the local topographic

wetness index, $\overline{TWI}_j$ the average TWI of all 25m×25m cells embedded in the TerrSysMP $j$ cell, $a_i$ is the drainage area upstream of cell $i$, and $\beta_i$ is the local slope angle [rad].

From the downscaled water table depth, we compute the water-saturated layer in the soil, i.e. the water pressure head term in Eq. 1 $h = d + wtd$, with $d$ computed with the soil thickness models presented in Section 2.1.2. Whenever the disaggregated water table is above the soil surface ($h > d$), then we assume the soil to be completely wet ($h = d$), whenever it is below the

local estimate of soil depth ($h < 0$), we assume the soil to be completely dry ($h = 0$).

## 2.2 Probabilistic approach

### 2.2.1 Rainfall threshold curves

We combine landslide inventory data in Switzerland and a daily gridded dataset of rainfall to develop rainfall threshold curves following the method of Leonarduzzi et al. (2017). The historical landslides were collected in the Swiss flood and landslide

damage database (Swiss Federal Research Institute WSL, Hilker et al., 2009). This database contains floods, landslides, and





rockfall events which produced damages in Switzerland since 1972. We select the landslide events that had a known location and date, and were not associated with snowmelt, for a total of 1807 events between 1981 and 2016 (timeframe of the analysis). The rainfall record is obtained as the interpolation of a network of ca. 430-460 raingauges, using the local climatology and regional precipitation-topography relationships (Shepard, 1984; Frei and Schär, 1998; Frei et al., 2006). It contains daily rainfall
totals on a 1km×1km grid covering the entire country since 1961.

For the definition of rainfall events, we follow the procedure introduced in Leonarduzzi et al. (2017). First we select susceptible cells, that is rainfall cells (1km×1km grid cells) where at least one landslide was recorded. For those cells we separate the rainfall timeseries into events, where an event is defined as a series of consecutive rainy days with a minimum of 1 dry day in-between events. These events are then classified as observed triggering if a landslide was recorded during or immediately
after them, and non-triggering otherwise.

We then define a power-law total rainfall-duration (ED) threshold curve $E = aD^b$ that objectively separates triggering and non-triggering rainfall events. To this end, we estimate the $a$ and $b$ parameters of the power law curve by maximising the True Skill Statistic (TSS=True Positive Ratio - False Positive Ratio), as in Leonarduzzi et al. (2017). This allows us to classify the rainfall events into the following groups by the calibrated ED threshold (see also Leonarduzzi and Molnar, 2020): observed
and correctly predicted triggering events above the ED curve (True Positives), observed triggering events which fall below the ED curve (Misses), observed non-triggering events which fall above the ED curve (False Alarms), and observed non-triggering events which fall below the ED curve (True Negatives).

### 2.2.2 Antecedent soil saturation

We use the values of soil saturation estimated by the Swiss operational hydrological model PREVAH (Viviroli et al., 2009)
at a 500m×500m resolution to check the added value of antecedent soil saturation on the ED curve predictions. PREVAH is a conceptual model, where the soil is represented by three storage modules: soil moisture storage (SSM), upper zone (unsaturated) runoff storage (SUZ), and lower zone (saturated) runoff storage. We use the values of the first two (unsaturated) layers and combine and transform them into a 0-1 soil saturation estimate. This is computed as: $soil saturation = (SSM + SUZ)/(SSM_{max} + SUZ_{max})$, where $SUZ_{max}$ is a distributed calibrated parameter, while $SSM_{max}$ is the maxi-
mum value of SSM simulated over the entire timeframe at each grid cell.

For each susceptible cell defined in Section 2.2.1, we extract the timeseries of the PREVAH soil saturation estimate at the corresponding cell and compute (a) the probability of the saturation to be smaller or equal to the saturation on the landsliding day(s), and (b) the average saturation during triggering and non-triggering events. This separation is used to analyse if soil saturation contains a predictive skill (signal) useful for landslide forecasting, i.e. if saturation is exceptionally high prior to
landsliding.

Finally, we test the information content of soil saturation for the ED curves, i.e. analyse whether information on soil saturation could reduce some of the misses and false alarms generated by the ED threshold curve estimated in Section 2.2.1. For each group of events (misses, false alarms, true positives and true negatives) and each rainfall event duration (1 to 6 days) we compute the mean soil saturation over 5-60 days prior to the beginning of the event. This allows us to examine if we fail





to predict some triggering events (Misses) with the ED curve because saturation was very high, reducing the rainfall amount required for the initiation of a landslide, and likewise if some larger rainfall amounts were insufficient for the triggering (False Alarms) because the soil was very dry before them. We also split rainfall events according to a saturation threshold and define two different optimum ED curves for generally dry and wet conditions. Such curves could be used in a landslide warning context together with estimated current wetness and forecasted rainfall.

## 3   Results

### 3.1   Physically-based approach

#### 3.1.1   Infinite slope model spatial patterns

Distributed inputs (Fig. 1) were used to compute the FoS across Switzerland as a function of the hydrological term $h$ for the two end-member states $h = 0$ and $h = d$. To do this, we first generated the distributed soil depth values following the four approaches introduced in Section 2.1.2 (Figure 2). These result in quite different spatial soil distributions, with the elevation-dependent soil depth mirroring the DEM, the slope-dependent soil depth showing low variability in depth in valleys and lowlands where slope is constant, and the linear diffusion model soil depth showing the highest spatial heterogeneity, with large differences in soil depth over short distances. This is due to the dependence on the second derivative of elevation (curvature), and results in low soil depth on mountain ridges, but sometimes larger values in convergent topography right next to them.

We then compute the minimum (assuming soil completely wet, $h = d$ in Eq. 1) and maximum (assuming soil completely dry, $h = 0$ in Eq. 1) FoS for every 25m×25m cell in Switzerland considering all four depth maps. We group the cells as unconditionally stable (when $FoS_{min} > 1$) and unstable (when $FoS_{max} < 1$), and conditionally (un)stable (all remaining cells) (Figure 3). The resulting limits of the FoS over the country seem not to be affected strongly by the soil depth model chosen. This is confirmed also when looking at the fraction of cells or landslides in each condition (Table 1). Nevertheless this is not to say that soil depth is not an important parameter for the initiation of landslides, as in Figure 3 and Table 1 we are not considering the interplay of soil depth and hydrology. Considering the two extreme scenarios (soil completely wet or completely dry), we are ignoring how likely these conditions are to occur, and the fact that a thicker soil will likely be more difficult to saturate. Therefore, while soil depth does not seem to impact the limiting conditions for completely wet and dry soil, it will very likely impact the landslide volume and the actual hydrological state and therefore FoS value.

Under the conditions studied here, only 22-25% of the area of Switzerland is conditionally unstable, i.e. area where hydrology matters for landslide occurrence according to the infinite slope model. The presence of so many landslides in unconditionally stable areas (65-66% of the total number of landslides), and the existence of some unconditionally unstable cells (10-13% of the country), are undesirable outcomes. While some inaccuracy in the location of the landslides (which might not refer to the detachment zone) could play a role, these results also suggest that either the infinite slope model is inadequate or the input parameters are inaccurate. In fact, the sensitivity of the FoS to cohesion makes the point (Figure 3 and Table 1) regarding parameter uncertainty. If we remove cohesion ($c = 0$), a much larger portion of the country is now susceptible to landslides

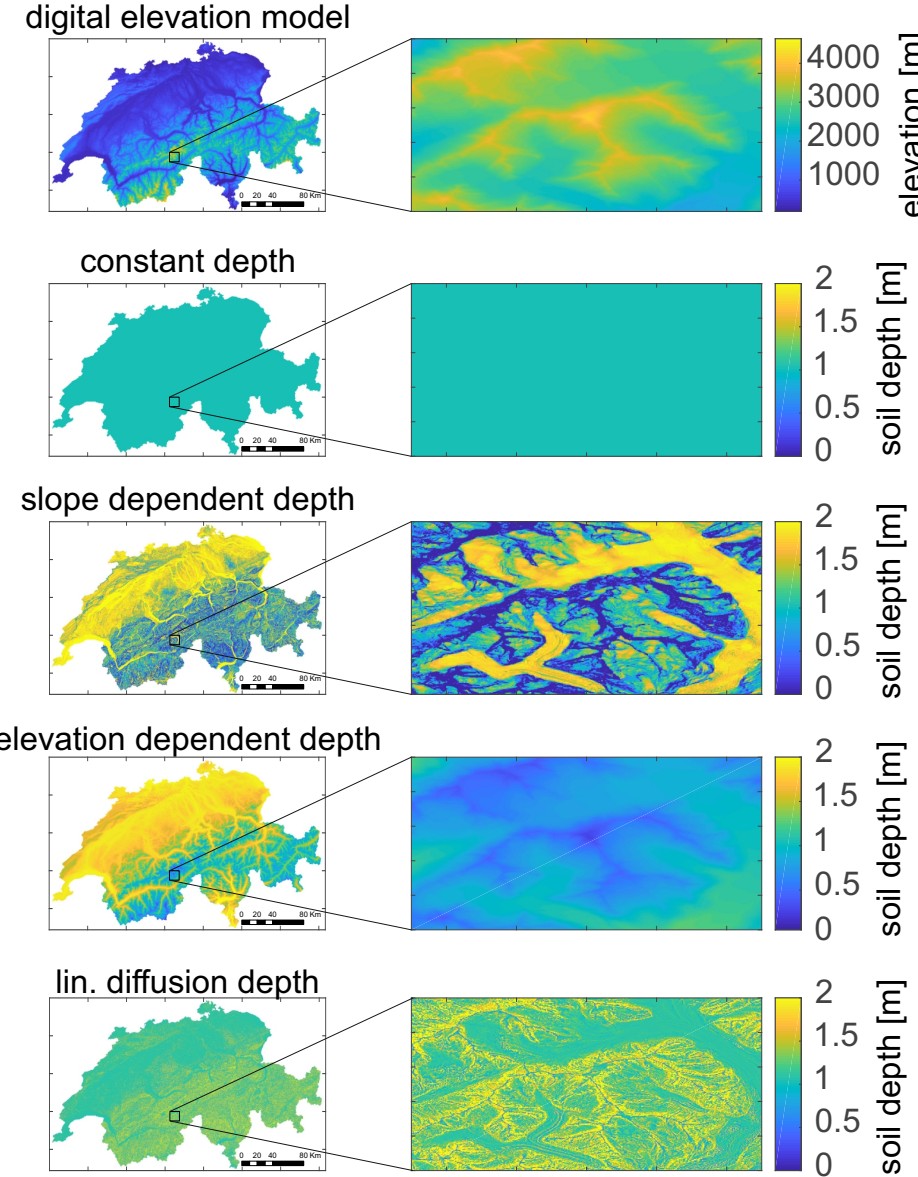

**Figure 2.** From top to bottom: digital elevation model (DEM, Swisstopo 25m) and 4 soil depth distributions: constant ($d$=1m), slope- and elevation-dependent, and $d$ between 0 and 2m assuming the linear diffusion model at steady state. The maps on the right side show a zoom of the area to appreciate small scale variability.

(unstable or potentially unstable), and the hydrologically active portion, conditionally (un)stable, is now 40% of the country, with more than 60% of the total landslides, and only 19% of the landslides remains in unconditionally unstable areas (3 and Table 1). This is a strong indication that the infinite slope model predictions are highly sensitive to input parameters. These
250   aspects and potential limitations of the FoS will be further discussed in Section 4.



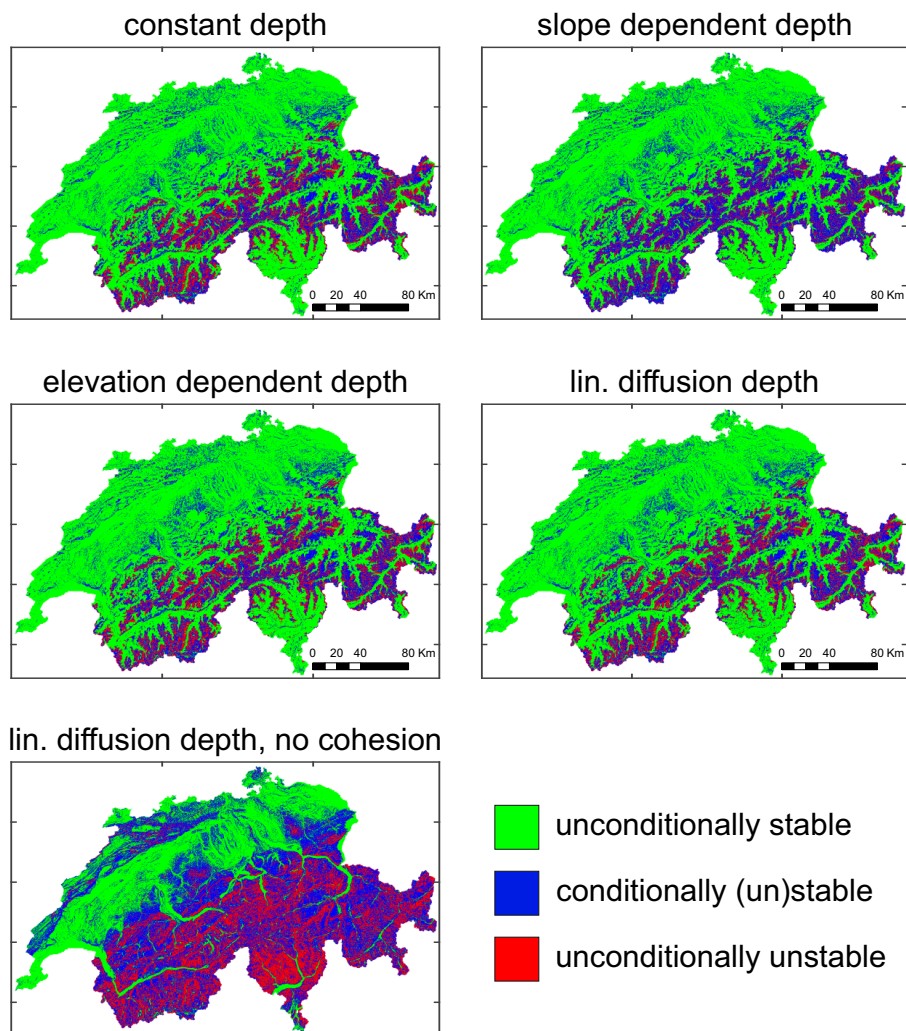

**Figure 3.** Maps of (un)conditionally (un)stable regions of Switzerland obtained from the two Factor of Safety limiting cases (soil completely wet or dry) and the different soil depth models. Panel in the bottom row is the reference case obtained with the linear diffusion model neglecting cohesion ($c = 0$).

### 3.1.2 The effect of dynamic hydrology

To address the temporal dynamics of FoS in the susceptible areas (25m×25m cells where at least one landslide was recorded) and their connection to observed landslides, we extract the daily timeseries of simulated TerrSysMP water table depth for all TerrSysMP cells within Switzerland for the period 1989-2018, and then compute the FoS in time for all 25m×25m cells in which at least one landslide was recorded by using the TWI downscaling.





**Table 1.** Percentage of Unconditionally Stable (US), Conditionally (Un)Stable (CUS), and Unconditionally Unstable (UU) cells in Switzerland according to the FoS calculations for each soil depth model, and percentage of landslides in each condition from landslide inventory. For the linear diffusion model the results are also shown when the cohesion is neglected ($c = 0$).

| soil depth model | US | CUS | UU | landslides in US | in CUS | in UU |
|---|---|---|---|---|---|---|
| constant | 66% | 22% | 12% | 65% | 30% | 5% |
| slope dep. | 66% | 25% | 10% | 65% | 30% | 5% |
| elevation dep. | 65% | 22% | 13% | 64% | 31% | 5% |
| lin. diff. | 65% | 22% | 13% | 64% | 31% | 5% |
| lin. diff. (no cohesion) | 35% | 40% | 25% | 19% | 61% | 20% |

The expectation is that landslides should occur when the FoS<1. While the value of 1 is often chosen as a theoretical threshold based on the balance of forces in a soil, several studies actually calibrate either the threshold FoS value or the critical area over which FoS<1 in a region (e.g., Casadei et al., 2003). In this work we accept that the critical value FoS=1 can vary spatially depending on the soil parameters and the performance of the hydrological model, and instead we look at the likelihood of landslide-triggering FoS values in individual cells from long-term simulations. To this end, we compute the probability of the FoS being equal or smaller than the value on the day a landslide was observed in the corresponding cell from long-term simulations and repeat this for all landslide cells (Figure 4). This probability should be small, showing that landslides tend to occur when the FoS is smaller in a given susceptible cell, regardless of the actual value, which might be location dependent.

Instead we observe that the probability of the FoS being equal or smaller than the triggering value seems to be small in only few cases (upper panel in Figure 4). Considering rainfall triggering potential of events as a ratio between triggering intensity and mean daily precipitation (Rtrig/mdp), which could help distinguish the events that are extreme, also does not show any signal (middle panel in Figure 4). There are many cases where the non-exceedance probability of triggering FoS equal to 1, which means that a very high FoS led to landsliding, which is contrary to expectations. However, these are also the cases in which the difference between mean FoS and triggering FoS is 0 (lower panel in Figure 4) and the standard deviation is also very small. This means that in those cells (i.e. the majority of cells), the simulated water table is constant in time or is always deeper than the soil thickness.

Similar conclusions can be drawn by looking at the FoS distributions during triggering and non triggering rainfall events (Figure 5). Once again we would expect that during triggering rainfall events the FoS is smaller. While this is indeed what we observe, the differences are barely noticeable, showing that this hydrological model-based simulation of $h$ would not be successful in a predictive setup in conjunction with an infinite slope model. In summary, adding a coarse resolution global hydrological model prediction of groundwater table to the FoS, even though it is downscaled to the higher spatial resolutions, reduces even further the hydrologically dynamic range of FoS predictions by producing constant and/or deep groundwater tables in our domain, and does not discriminate well between triggering and non-triggering conditions at the scale of individual landslides.





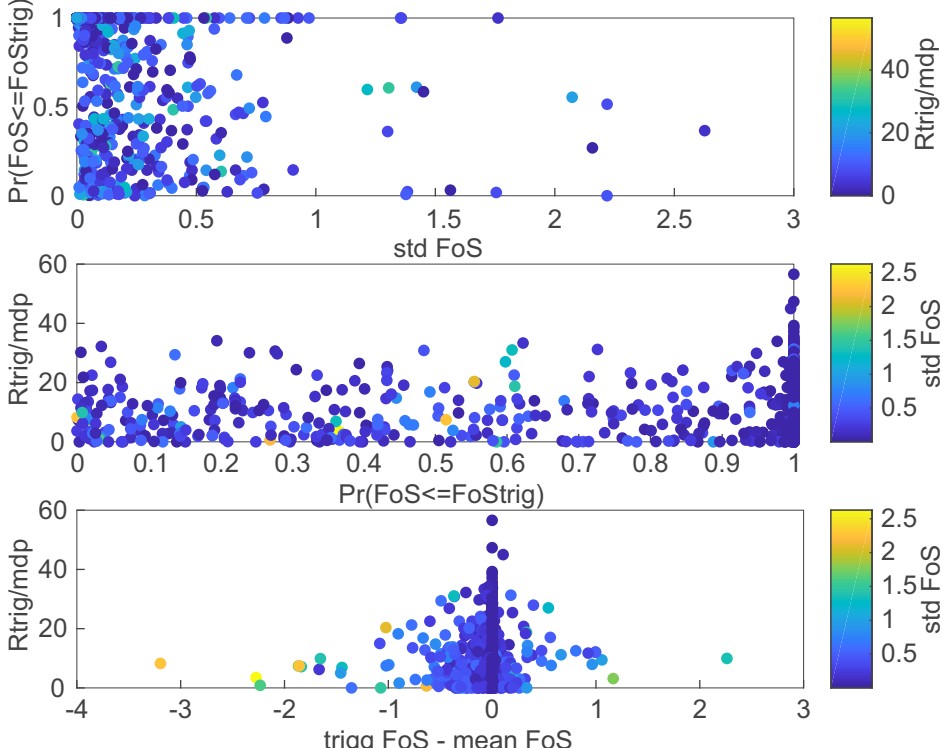

**Figure 4.** Scatter-plot of different combinations of triggering rainfall and Factor of Safety (FoS) for all landslides (each point corresponds to a landslide): the probability of the FoS in the cell being smaller than the value on the day of the landslide (Pr(FoS<=FoStrig)), the standard deviation in time of the FoS in the cell (std FoS), the ratio between the triggering rainfall intensity (rainfall intensity on the day of the landslide) and the cell mean daily precipitation (Rtrig/mdp), and the difference between the triggering FoS and the temporal mean FoS of the cell (trigg FoS - mean FoS).

## 3.2 Probabilistic approach

The role of antecedent wetness and the information content of the saturation estimates provided by the hydrological model PREVAH (Viviroli et al., 2009) for landslide prediction is tested in a similar way to the physically-based approach, but because we do not have an estimate of water table depth in PREVAH, we use the soil saturation instead. We expect patterns opposite to that of the FoS: the saturation to be exceptionally large on landslide days and generally larger during triggering than non-triggering events. This is exactly what we observe considering the probability of the saturation being smaller or equal to that of the landslide day (Figure 6) and the distribution of average saturation during triggering and non-triggering events (Figure 7). The probability of a cell being wet at the time of landsliding is very high in most susceptible cells (upper and middle panel in Figure 6). Also the average saturation during triggering rainfall events is clearly larger than during non-triggering events, especially the maximum saturation reached (Figure 7).





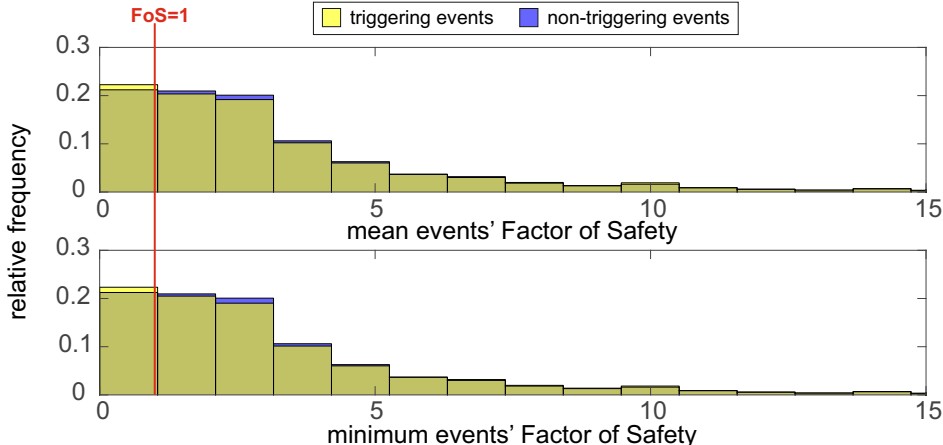

**Figure 5.** Histograms of the mean (top) and minimum (bottom) Factor of Safety during triggering and non-triggering rainfall events, combining spatial (i.e. differences between landslide locations) and temporal (i.e. differences between events in the cells) differences.

These results suggest that the saturation estimate provided by PREVAH, might contain information useful for the prediction of landslides. While no significant signal in precipitation combined with soil saturation, i.e. triggering rainfall being larger for events with smaller antecedent soil saturation, could be identified in the previous results for individual events (Figure 6), we choose to combine saturation estimates with ED thresholds for landslide prediction. We first define the optimum ED threshold for landsliding by maximising the TSS ($E = 20.1D^{0.74}$, TSS=0.656), and then compute the average saturation for each duration and class of events: false alarms, misses, true positives, and true negatives. If saturation could improve the ED threshold predictions it would reduce the number of misses and false alarms. Indeed this is what we observe in the mean antecedent saturation trends: regardless of the number days prior to the beginning of the rainfall event over which the mean saturation is computed, the misses (T below in Figure 8) are always associated with the highest antecedent saturation, and the false alarms (NT above in Figure 8) with the lowest saturation. This confirms that at least some of the misses were triggered by a smaller rainfall amount than expected due to exceptionally high antecedent soil wetness, whereas sometimes although the ED threshold was exceeded, no landslide event was observed due to exceptionally low saturation prior to the rainfall event.

While generating hydrometereological thresholds (rainfall-saturation thresholds) does not lead to improved performances compared to the ED threshold, separating the events depending on the antecedent saturation and then optimising separately two ED thresholds does. We obtain a slightly larger threshold ($E = 22.1D^{0.76}$) for events with antecedent 5 days average saturation below 0.3 than for those with prior saturation antecedent wetness than 0.3 ($E = 18.1D^{0.74}$), confirming that when the soil is drier prior to the rainfall event, more rainfall is required to trigger a landslide. These combined thresholds lead to an overall TSS of 0.664, so a small improvement in model predictions.

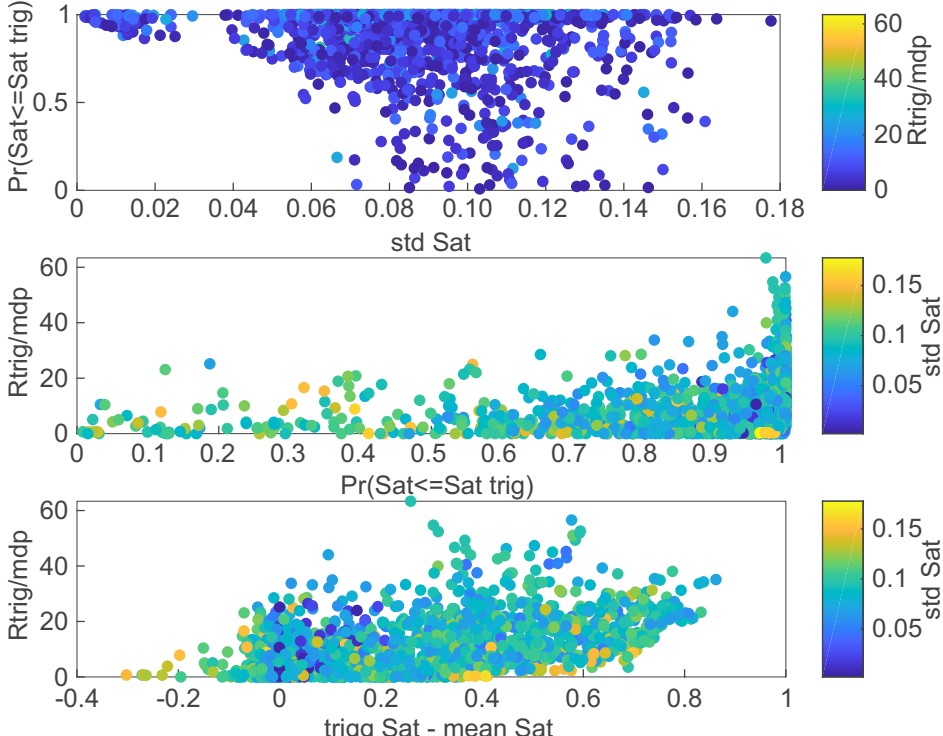

**Figure 6.** Scatter-plot of different combinations of the rainfall and saturation (from the hydrological model PREVAH) properties for all landslides (each point corresponds to a landslide): the probability of the saturation in the cell being smaller than the value on the day of the landslide (Pr(Sat<=Sat trig)), the standard deviation in time of the saturation in the cell (std Sat), the ratio between the triggering rainfall intensity (rainfall intensity on the day of the landslide) and the cell mean daily precipitation (Rtrig/mdp), and the difference between the triggering saturation and the temporal mean saturation of the cell (trigg Sat - mean Sat).

## 4 Discussion

The results presented here suggest that the probabilistic approach with rainfall and saturation thresholds is superior to the

physically-based approach with the Factor of Safety calculation. It is important to stress that this is not a general conclusion, but rather a conclusion drawn from the specific models which we compared. In fact, if a physically-based approach would accurately capture the pore water pressure variations at the required high resolution scales and therefore reproduce and predict slope failure with the FoS (or another geotechnical) model, we maintain that it would be superior to a probabilistic approach. It is therefore worthwhile to discuss the limitations of the tested physically-based approach and the results obtained with regards

to the geotechnical component (i.e. the infinite slope approach and FoS calculations) and those related to the hydrological component.

To look at the infinite slope approach independently from the hydrology, we can focus on the analysis of conditionally and unconditionally stable/unstable areas of the country and their validation against the location of historical landslides. There are





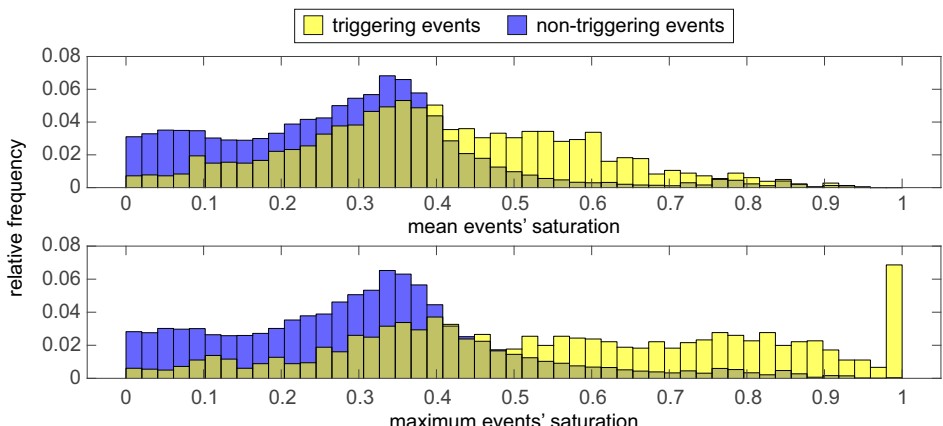

**Figure 7.** Histograms of the mean (top) and maximum (bottom) saturation estimated by the hydrological model PREVAH during triggering and non-triggering rainfall events, combining spatial (i.e. differences between landslide locations) and temporal (i.e. differences between events in the cells) differences.

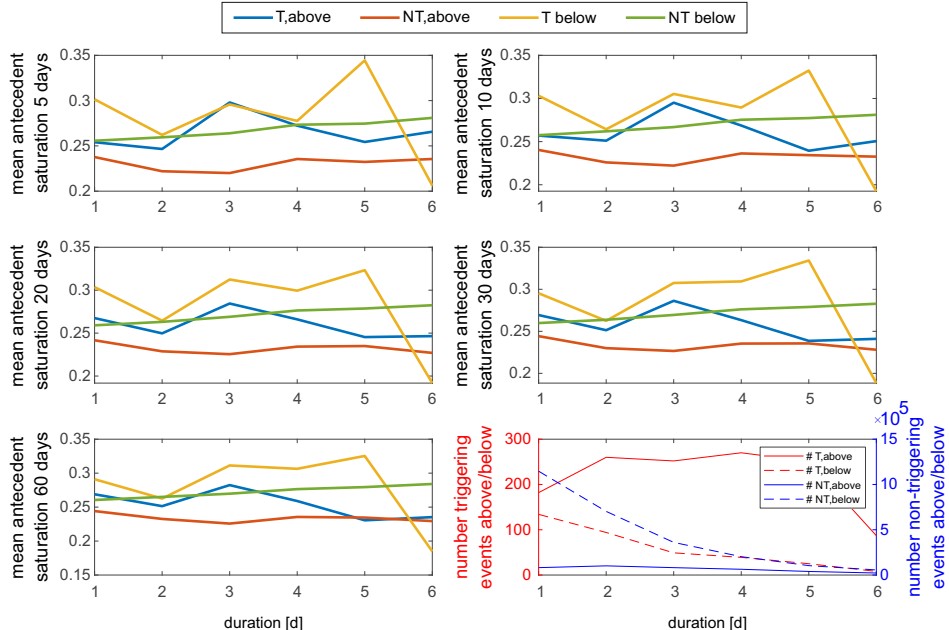

**Figure 8.** Plots of mean antecedent saturation averaged over 5-10-20-30-60 days prior to the beginning of the corresponding rainfall event for durations of 1-6 days. Events are divided into 4 groups: true positives (Triggering, above), false alarms (Non-Triggering, above), misses (Triggering, below), and true negatives (Non-Triggering, below). The plot in the lower right shows the number of events in each group of events for each duration, to check the robustness of the mean estimates.





two concerning aspects in these results: the presence of so many historical landslides (65-66%) in unconditionally stable areas

and the existence of unconditionally unstable areas. The uncertainty in the location of the landslides could explain some of the slope failures in unconditionally stable areas. Out of the 1354 landslides in unconditionally stable (US) areas, for 937 there are no US cells in the 24 neighbouring cells (area of 125m×125m centred on the cell), and for 739 not even in the 80 cell neighbourhood (area of 275m×275m centred on the cell). Furthermore, unconditionally unstable areas should theoretically not exist, because they should have failed already or have 0 soil depth. Nevertheless roughly 10-13% of the country is classified as

such. These two outcomes are therefore failures of the infinite slope approach. This approach is based on strong simplifications and ignores potentially important processes such as suction in unsaturated soils, which increases stability. Nevertheless, we believe believe the uncertainties in the input parameters have the strongest influence. Proof of this are the results obtained neglecting vegetation cohesion (Figure 3). The fact that the map of (un)conditional (in)stability changes considerably when removing cohesion, shows the sensitivity of the FoS calculations to input parameters. Other input parameters may be similarly

influential. For instance, the friction angle values obtained based on the the soil texture map from OpenLandMap (Figure 1b), are practically homogeneous over the country. We expect that this property is in reality much more heterogeneous and, together with cohesion, is affecting the unconditionally stable area. The sensitivity of the FoS estimates to the uncertain parameters can be examined by Monte Carlo simulations, provided that parameters distributions are known (Hammond et al., 1992; Pack et al., 1998; Griffiths et al., 2011). Soil depth is also a very uncertain and influential parameter. The UU areas, in the alpine region,

are very likely steep locations were the soil is absent (exposed bedrock). This aspect is missed by most soil datasets as well as soil depth models.

Another important aspect to consider for the FoS calculation is the spatial resolution. Higher resolutions allow to better capture the local heterogeneities (if data is available), most importantly the topography (i.e. slope). On the other hand, at high resolutions, the assumption of slope length much greater than soil depth becomes invalid and if the cell size becomes much

smaller than the typical detachment area of landslides, the interactions between neighbouring cells become even more critical. For this reason, several geotechnical models have been developed that explicitly model progressive failure, lateral interactions and stress redistribution (Cohen et al., 2009; von Ruette et al., 2013; Anagnostopoulos et al., 2015; Fan et al., 2015).

The limitations of the hydrological component regardless of the geotechnical model, are evident from the very small information content in the water table depth estimation produced by TerrSysMP. In our analysis we focused on the temporal

variability of the FoS (i.e. difference in time relative to the mean for each susceptible cell, or comparison of mean maximum and minimum FoS during triggering and non-triggering events), and the only variable in the FoS calculation which can vary in time is the soil water pressure. This means that the lack of temporal variability in the FoS is a direct consequence of the lack of temporal variability in the water pressure head. The hydrological estimates from the physically-based framework wouldn't be successful even if used with rainfall in hydrometereological thresholds, regardless of the geotechnical component. The signal

is instead evident when considering soil saturation obtained from PREVAH. Theoretically, a physically-based model should be better capable of simulating the movement of water in the soil and therefore predicting the saturation or water table depth more accurately.





We believe these results are a direct consequence of the resolution. In fact, at such coarse resolution, the model is simulating the large scale fluctuations in ground water depth, which are indeed generally very small, and does not capture the local
changes, driven by higher resolution topography. The downscaling with the topographic wetness approach cannot compensate for this. While it can account for local topography by lowering/raising the water table depth, this correction is constant in time. Therefore, if the coarse hydrological variable does not include enough temporal variability, so will the higher resolution downscaled estimate.

For the specific cases presented here, having a higher spatial resolution (500m×500m rather than 12.5km×12.5km) in a
conceptual hydrological model is more beneficial than the gain in accurate physical representation of the processes. This stresses once more the importance of adequate spatial resolution of hydrological models, especially for the assessment of slope and soil saturation dependent natural hazards such as landslides.

## 5  Conclusions

We explore two approaches for the prediction of landslides and the value of soil wetness in these predictions applied to a
regional scale case study in Switzerland. In the first approach we use the water table depth estimates from a coarse-resolution physically-based model (TerrSysMP) dowscaled by the topographic wetness approach and slope stability assessment using the infinite slope approach. In the second appraoch we use rainfall-duration threshold curves informed by soil saturation obtained by a high resolution conceptual hydrological model (PREVAH).

Our main findings are:

– the infinite slope approach for quantifying slope instability is largely affected by the accuracy of input soil parameters, in particular cohesion in our case (removing cohesion doubled the area where hydrology mattered in FoS prediction)

– soil depth does not seem to affect the estimate of (un)conditionally (un)stable areas, although it is an essential parameters for the estimate of local wetness and determines the landslide volume

– according to the infinite slope approach and the parameters considered here, hydrology can play a role in the initiation
of landslides over only ca. 20% of Switzerland (the conditionally (un)stable area, where about 30% of all observed landslides have occurred)

– saturation estimates from a high resolution conceptual hydrological model (PREVAH) are useful in improving landslide predictions based on rainfall only and can be exploited in combined saturation and ED thresholds

– while we maintain the best approach would be physically-based modelling of hydrology combined with geotechnical
modelling, resolution is a very important aspect and a semi-conceptual model at a high resolution (500m×500m) provides more dynamic and useful estimates of the soil status than a physically-based hydrological model run at a much coarser resolution (12.5km×12.5km) despite static downscaling.



*Data availability.* The friction angle were obtained from Geotechdata.info (Angle of Friction, http://geotechdata.info/parameter/angle-of-friction.html, as of September 14.12.2013, last accessed 07.07.2020). All other soil maps were downloaded from the OpenLandMap website

(www.openlandmap.org, last accessed 30.04.2020). The 25m digital elevation model was provided by swisstopo (https://shop.swisstopo.admin.ch/en/products/height_models/dhm25). The rainfall products were provided by the Swiss Federal Office of Meteorology and Climatology MeteoSwiss (available for research purposes upon request). The Swiss Federal Research Institute WSL provided the landslide data (available for research purposes upon request) and the PREVAH simulation results. TerrSysMP hydrological simulation results were downloaded from https://datapub.fz-juelich.de/slts/cordex/index.html (last access 16.06.2020, DOI: http://doi.org/10.17616/R31NJMGR)

*Author contributions.* E. Leonarduzzi conducted the analysis. E. Leonarduzzi and P. Molnar conceived the research. All authors contributed to writing the paper.

*Competing interests.* The authors declare they have no competing interest





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
