# Peer review of "Rainfall-induced shallow landslides and soil wetness: comparison of physically based and probabilistic predictions"

_Hydrology and Earth System Sciences, 2020_

## Referee Comment (RC1) · Anonymous Referee #1 · 7 Feb 2021

General comments: The topic of this study presented in this paper is interesting. The manuscript attempts to present the comparison of two different methods applied for rainfall-induced shallow landslide prediction. However, the data used for each method are mainly obtained from the database and estimations. So, it is hard to see the novelty of the study presented in this paper. The presentation of the results is quite difficult to understand since the authors presented the results of the probabilistic approach in graphical forms. The discussion covered something that has already been understood from the results of using data from estimations. The manuscript is more like a technical paper than an academic paper in the present format. This paper also needs to be grammatically corrected.

Specific comments: 1. Introduction Provide some reasons/ justifications for selecting the physical-based model and probabilistic in your study. Please state the urgency of comparing these two methods. What was the hypothetical background that prompted you to compare these two methods?

2. Methods: (a) The physical-based modeling is dependent upon the accuracy of soil properties data. Your current modeling study used estimated soil properties. Please state how accurate your estimation of soil unit weight, cohesion, and friction angles using method/ approach. Please provide some justifications of these approaches/ methods, perhaps by presenting some previous studies results in the Introduction section. (b) All datasets OpenLandMap are at a resolution of ca 250x 250 m, but the modelling considered a resolution of DEM at grid cell size of 25x25 m. Please state what method you chose to resolve this difference in map resolution. (c) Please justify choosing the values of cohesion due to tree roots of 5 - 22 kPa. The values may indicate that the soil layers have low strength.

3. Results The results of the physical modeling and probabilistic approaches also need to be presented in a spatial format and then validated with historical landslides.

4. Conclusions: please check again whether the conclusions drawn have answered all the research questions. You seemed to miss answering questions 1 and 3 stated in the Introduction section.

5. Please check the writing again for grammatical errors (see the attached file). You can use Grammarly to find the errors and to get suggestions for the corrections.

Please also note the supplement to this comment: https://hess.copernicus.org/preprints/hess-2020-624/hess-2020-624-RC1-supplement.pdf

---

## Referee Comment (RC2) · Anonymous Referee #2 · 8 Feb 2021

The paper investigates the use of (very) large scale hydrological modeling to improve the prediction of shallow landslide occurrence throughout Switzerland, compared to the typical approach based on the statistical analysis of triggering precipitation alone. This is a quite actual topic, which can be of interest for the readership of HESS, as most landslide early warning systems still rely only on precipitation information, while there is physical and operational evidence that including hydrologic information may be useful in many cases.

The paper is well organized and the English language pretty good, although the choice of the scatter plots adopted for presenting the results may result a bit awkward (al-
though quite synoptic, which is a good point). However, I find some issues in the adopted modeling approaches, which somehow affect also the results and the conclusions, so I believe that major revisions are needed before reevaluating the manuscript for possible publication.

Specifically, all the paper deals with the comparison of a "physically-based" hydrological model, run over a coarse spatial grid and coupled with a simplified slope equilibrium equation based on the infinite slope hypothesis so to end with an assessment of a safety factor value at any point of the grid, with a purely probabilistic evaluation of the coupled effects of slope conditions in landslide and non-landslide days, carried out by estimating soil degree of saturation with a conceptual hydrological model run at a much finer spatial scale. The obtained results indicate that the "physically-based" approach is largely outperformed by the probabilistic, and the discussion ascribes this outcome mostly to issues related to the coarse resolution (i.e. wrong local estimates of soil depth, slope inclination, soil mechanical properties, slope hydraulic response to precipitations, and so on). All these discussion points are clearly valid and acceptable, but I believe that the Authors should more deeply describe, discuss and comment the limitations of the model that they consider as "physically-based".

I understand that the Authors probably mean that with such a modeling approach they assess landslide occurrence with an equilibrium equation, that is the application of a physical principle. However, although no detail is provided about the characteristics of the infiltration model which provides the water table depth for the application of the equilibrium equation, I have the feeling that it may be not completely physically based. In fact, while dealing with shallow landslides, which occur in initially unsaturated soil covers (as the Authors indeed notice in the Discussion section), it is assumed that the infiltration process results in the building of a water table at some depth in the soil, which is not necessarily the case (it strongly depends on the assumed boundary condition at the base of the soil cover), and which seems more a conceptualization of the effects of infiltration, rather than the result of a physically-based model of rainfall infiltration process. The adopted expression (1) of the factor of safety (and the obtained results, as well as the discussion about them) seem to be deeply affected by this conceptualization.

1. When the soil is considered dry and cohesionless (h=0 in equation (1)), FoS reduces to tan(fi)/tan(beta), which implies that soil depth is ineffective and that everything depends on the quality of your topographic map (beta varies much more than fi, which not surprisingly remains always not far from 30° for the kind of soils you may have in mountain environment).

2. When root-cohesion is introduced (by the way, another conceptualization), considering dry slopes with inclination larger than 30°, you can easily see that it mostly results FoS>tan(fi)/tan(beta)+0.1c, so that even with the smallest hypothesized cohesion (5 kPa), FoS can be smaller than one only for slopes more inclined than 50°.

3. When saturated soil cover (h=d) is considered without root cohesion, it is FoS=(g-gw)/g*tan(fi)/tan(beta), that, with the values of g that you assume for the soil (seemingly between 1.2 and 1.6, with gwïĄĂ1) leads to stability possible only for inclinations smaller than 12°, independent of soil depth.

4. If we introduce root cohesion when h=d, you get again that for slopes with inclination above 30° some 0.1c is summed up to the previous expression of FoS, that is FoSïĄĂ0.1c+(g-gw)/g*tan(fi)/tan(beta), so that only when cohesion is the smallest (c=5 kPa) you may get some slope inclinations for which stability depends on the value of the water table h.

This given, my overall impression is that all the results from the "physically-based" modeling, namely all the considerations about (un)conditional (un)stable situations, and their comparison with the landslide inventory are strongly affected by the weakness of the model, before than by the issues related to the coarse modeling grid. I mention a few points that I believe are worth some discussion: (i) what is the meaning of gamma in equation (1)? This value should change according to soil saturation, and

the assumed values between 12 and 16 kN/m3 seem rather to refer to some average field condition (this certainly has an effect on the predicted values of FoS); (ii) to what extent the assumption of the building of a water table is acceptable and consistent with the geomorphological characteristics of the studied alpine slopes (i.e. type of soil and type of bedrock)? (iii) is groundwater table (likely much deeper than the shallow soil covers of interest for the study, as the Authors themselves observe at lines 270-271) an appropriate variable to be chosen for the purpose of this study about shallow landslides? (iv) I guess that TerrSysMP model offers also soil moisture data, so why did you choose groundwater table for your analyses?

Concluding on this point, I still believe that the attempt to exploit the information available from a model like TerrSysMP for the sake of predicting landslides is a valuable task, and that it merits to be investigated. But it seems to me that this could be made with more care than it is in this study.

On the other hand, there is the conceptual hydrologic model and the use of estimated soil moisture with a probabilistic approach to improve landslide assessment carried out with empirical precipitation thresholds. While this part is more straightforward, there is still a major point that should be clarified. Your aim is to investigate the potential of soil moisture prior the onset of triggering rainfall to improve empirical thresholds. Despite this, from figures 6 and 7 it seems that you never consider this information, as only saturation on the day of the landslide, maximum or mean saturation during an event, and general statistics of the saturation in the cells are calculated. The discussion of the moisture conditions prior the event is limited to graphs of fig. 8, considering mean saturation for 5-60 days long periods preceding rainfall events. Some discussion of the graphs would be worth. For instance: the 5 and 10 days averages seem to be the best choice to correct false alarms (red line well below the others); long events (6 days) seem to lose memory of the effects of initial conditions on missed alarms (all yellow lines drop down for 6 days, while they are above all other lines for shorter event durations). Instead, in the paper only the brief sentence at lines 303-304 is

dedicated to the possibility of building hydrometeorological thresholds, which are just said to be uncapable of improving the performance of precipitation thresholds without any information. I think that much more discussion and data should be presented to the reader, as the effects of prior soil moisture is all in all the focus of the paper.

In addition to these two major issues, you can find some remarks and comments as annotations in the attached file.

Please also note the supplement to this comment:
https://hess.copernicus.org/preprints/hess-2020-624/hess-2020-624-RC2-supplement.pdf

[Figure]

**Supplement:**

[revised manuscript text omitted]

---

## Author Comment (AC1) · 12 Mar 2021

We thank the reviewer for the review and the constructive comments. We address here all the points raised, and we indicate how we will take care of them in the revision.

*General comments: The topic of this study presented in this paper is interesting. The manuscript attempts to present the comparison of two different methods applied for rainfall-induced shallow landslide prediction. However, the data used for each method are mainly obtained from the database and estimations. So, it is hard to see the novelty of the study presented in this paper.*

While the methodologies used in the manuscript are not novel, but rather taken from ours and other previous studies, the messages derived from the analyses and the comparisons carried out are in our opinion relevant, important, and new. Above all, the superiority of the soil wetness estimates from the conceptual higher resolution model highlights that the resolution of the hydrological model is critical, even more than the physical description behind the hydrological model (i.e., a conceptual model at 500m resolution is found to be superior to a physically-based one at 12.5 km resolution). This conclusion comes even before getting into the methodology for landslide prediction (i.e., physically based modeling of soil moisture and stability vs a probabilistic approach based on hydrometeorological thresholds).

*The presentation of the results is quite difficult to understand since the authors presented the results of the probabilistic approach in graphical forms.*

We are not sure we follow what the referee means by "graphical form" only. In the probabilistic approach first results just concerning the hydrological component are shown (Figure 6 and 7) and commented in lines 281-293, and in combination with rainfall thresholds in lines 293-301. Secondly, we utilize the soil wetness estimates in a probabilistic approach, by defining two rainfall thresholds depending on antecedent wetness. These thresholds are reported and commented in lines 302-307. The probabilistic approach only deals with the temporal dimension of landslide triggering, and only focuses on triggering conditions (rainfall and antecedent wetness), but not whether a landslide is actually possible at a given location (which is what susceptibility mapping does). Therefore, when considering a probabilistic approach one should look at the temporal dimension (i.e., whether wetness and rainfall conditions are exceptional when a landslide happened), which is better explained by figures such as Fig 7 and 8 than purely with skill metrics in tables.

*The discussion covered something that has already been understood from the results of using data from estimations.*

The discussion builds on the results presented in the previous sections and focuses on understanding and explaining the results obtained, mainly focusing on the "failure" of the physically based approach, which we did not anticipate at the start of the study. All the limitations of the proposed framework (i.e., methods and data) are analyzed, to identify the main sources of error.

*The manuscript is more like a technical paper than an academic paper in the present format. This paper also needs to be grammatically corrected.*

We believe that this manuscript is an academic paper as it doesn't just follow established methodologies, but combines and explores aspects from several different studies (i.e., probabilistic approach, infinite slope approach, TWI downscaling, soil depth estimation, physically based susceptibility mapping, etc.), providing a fair comparison of two approaches which not only stresses the

known importance of antecedent conditions, but also highlights what the limitations are of such approaches at the regional scale. Grammar corrections will be made where errors are found.

*Specific comments: 1. Introduction Provide some reasons/ justifications for selecting the physical-based model and probabilistic in your study. Please state the urgency of comparing these two methods. What was the hypothetical background that prompted you to compare these two methods?*

The probabilistic approach is the most commonly applied method at the regional scale in landslide predictions. Physically based approaches are often only applied at the local scale (e.g. in small watersheds where landslides occurred), due to computational and data limitations, but could in principle provide better predictions at any scale. The idea behind this study was to objectively compare these two methods at a large scale by choosing readily available hydrological estimates and data, and robust and widely used methodologies for this purpose (rainfall thresholds and infinite slope approach). We will explain the context and purpose of the comparison better in the revised manuscript.

*2. Methods: (a) The physical-based modeling is dependent upon the accuracy of soil properties data. Your current modeling study used estimated soil properties. Please state how accurate your estimation of soil unit weight, cohesion, and friction angles using method/ approach. Please provide some justifications of these approaches/ methods, perhaps by presenting some previous studies results in the Introduction section.*

The reviewer is correct in raising this point concerning parameter uncertainty. The uncertainty of the soil parameters is either directly coming from the uncertainty in the SoilGrids dataset in OpenLandMap, or a combination of that and other choices, such as in the case of friction angle, where the values are chosen from literature, given the soil classification. It is complex to quantify the combined uncertainty rigorously. Because we found that the limitation of the suggested physically based approach was the lack of temporal dynamics, which is absolutely independent from the infinite slope approach and its parameters, we decided not to specifically quantify the effects of soil parameter uncertainties, for example by Monte Carlo analysis. However, we did quantify the effect of the single most important uncertainty that is the soil depth, by scenario analysis with four different soil depth estimation methods typically used in scientific investigations and in practice.  We will expand the discussion of these uncertainties and their implications in the Discussion section of the revised manuscript.

*(b) All datasets OpenLandMap are at a resolution of ca 250x 250 m, but the modelling considered a resolution of DEM at grid cell size of 25x25 m. Please state what method you chose to resolve this difference in map resolution.*

For each 25m x 25m cell, the value of the 250m x 250 m in which the cell is completely or mostly contained is chosen. We will add this explanation in the revised manuscript. It should be recognized that the high resolution of the topography is primarily needed for a slope gradient estimation, which is closest to the appropriate scale of the infinite slope stability model. The coarser scale of the soil and landcover data are a secondary and less important problem.

*(c) Please justify choosing the values of cohesion due to tree roots of 5 - 22 kPa. The values may indicate that the soil layers have low strength.*

As already explained in the text, the range of cohesion values and the individual values assigned to each land cover class were decided based on previous studies that either provide measured ranges and

suggest that denser tree covers and mixed forests are associated with larger values of cohesion (Schwarz 2012, Dorren and Schwarz, 2016). We have attempted to cover a range of values of root added cohesion reported in studies in experiments (e.g. Cazzuffi et al., 2014). The referee is reminded that the added cohesion by roots is strongly depth dependent and for our estimation with the Mohr-Coulomb approach we need a depth integrated value for the soil layer, for which the range up to ca. 20 kPa is realistic for a range of herbaceous plants and tree species.

*3. Results The results of the physical modeling and probabilistic approaches also need to be presented in a spatial format and then validated with historical landslides.*

There must be a misunderstanding here. All the analyses are done considering the spatial dimension and the historical landslides from the database mentioned in the paper. All figures 4-8, which present the probabilistic and physically based approach are done comparing to the historical landslides. For each landslide location, the rainfall events and antecedent wetness conditions or factor of safety values are separated into triggering (if a landslide happened in that location during the rainfall event) or non-triggering otherwise. As mentioned in the second response above, the probabilistic approach doesn't deal at all with the spatial susceptibility (i.e., whether at a location landslides are to be expected or not), which is the explanation for the absence of spatial maps for this approach. In the physically based approach, the spatial dimension is considered through the infinite slope approach. This is shown in figure 3 with the wet and dry boundary scenarios and in Table 1, where the percentage of landslides is reported for each region (unconditionally stable/unstable and conditionally unstable). As mentioned in lines 257-259, adding dynamic hydrology, with the downscaled TerrSysMP estimates, reduces even more the conditionally (un)stable portion of the domain, as most cells are simulated constantly at saturation. We do not see what else we could add about the spatial analyses that would add more clarity.

*4. Conclusions: please check again whether the conclusions drawn have answered all the research questions. You seemed to miss answering questions 1 and 3 stated in the Introduction section.*

In the revised manuscript we will revise this so that conclusions and research question in the introduction will match closely.

*5. Please check the writing again for grammatical errors (see the attached file). You can use Grammarly to find the errors and to get suggestions for the corrections.*

Indeed, there were some small grammar errors in the text, but what the referee is pointing to are not grammar errors rather styles of writing. Two of the co-authors use English as a mother tongue, we will make sure that grammar errors are removed.

Cazzuffi, D., Cardile, G. and Gioffrè, D.: Geosynthetic Engineering and Vegetation Growth in Soil Reinforcement Applications. Transp. Infrastruct. Geotech. 1, 262–300 (2014). https://doi.org/10.1007/s40515-014-0016-1

Dorren, L. and Schwarz, M.: Quantifying the stabilizing effect of forests on shallow landslide-prone slopes, in: Ecosystem-Based Disaster Risk Reduction and Adaptation in Practice, pp. 255–270, Springer, 2016.

Schwarz, M., Cohen, D., and Or, D.: Spatial characterization of root reinforcement at stand scale: theory and case study, Geomorphology, 171, 190–200, 2012.

---

## Author Comment (AC2) · 12 Mar 2021

We thank the reviewer for the review and the constructive comments. We address here all the points raised, and we indicate how we will take care of them in the revision.

*The paper investigates the use of (very) large scale hydrological modeling to improve the prediction of shallow landslide occurrence throughout Switzerland, compared to the typical approach based on the statistical analysis of triggering precipitation alone. This is a quite actual topic, which can be of interest for the readership of HESS, as most landslide early warning systems still rely only on precipitation information, while there is physical and operational evidence that including hydrologic information may be useful in many cases.*

*The paper is well organized and the English language pretty good, although the choice of the scatter plots adopted for presenting the results may result a bit awkward (although quite synoptic, which is a good point).*

We agree that the scatter plots (e.g. Fig 4 and 6) are not easily readable. The idea was to look at how "exceptional" triggering days are in terms of antecedent (saturation or FoS) and triggering (rainfall) conditions. We will improve the description and consider simplifying them.

*However, I find some issues in the adopted modeling approaches, which somehow affect also the results and the conclusions, so I believe that major revisions are needed before reevaluating the manuscript for possible publication.*

*Specifically, all the paper deals with the comparison of a "physically-based" hydrological model, run over a coarse spatial grid and coupled with a simplified slope equilibrium equation based on the infinite slope hypothesis so to end with an assessment of a safety factor value at any point of the grid, with a purely probabilistic evaluation of the coupled effects of slope conditions in landslide and non-landslide days, carried out by estimating soil degree of saturation with a conceptual hydrological model run at a much finer spatial scale. The obtained results indicate that the "physically-based" approach is largely outperformed by the probabilistic, and the discussion ascribes this outcome mostly to issues related to the coarse resolution (i.e. wrong local estimates of soil depth, slope inclination, soil mechanical properties, slope hydraulic response to precipitations, and so on). All these discussion points are clearly valid and acceptable, but I believe that the Authors should more deeply describe, discuss and comment the limitations of the model that they consider as "physically-based".*

We refer to these approach as physically based according to the classical definition in this field, insofar hydrological and geotechnical modelling is concerned. We focused on the issues associated with the resolution, because they are definitely key here. In fact, from a process description point of view for surface and subsurface water fluxes, the physically-based framework of TerrSysMP should be clearly superior to the conceptual PREVAH, and the resolution must be a key difference. Solving water flow equations over a grid of 12.5 km is simply too coarse to get the fluxes right, regardless of the accuracy of the physics behind the flow equations. Nevertheless, there are of course other limitations also associated with all the components of the modeling framework, we will add a comment on these.

*I understand that the Authors probably mean that with such a modeling approach they assess landslide occurrence with an equilibrium equation, that is the application of a physical principle. However, although no detail is provided about the characteristics of the infiltration model which provides the water table depth for the application of the equilibrium equation, I have the feeling that it may be not completely physically based. In fact, while dealing with shallow landslides, which occur in initially unsaturated soil covers (as the Authors indeed notice in the Discussion section), it is*

*assumed that the infiltration process results in the building of a water table at some depth in the soil, which is not necessarily the case (it strongly depends on the assumed boundary condition at the base of the soil cover), and which seems more a conceptualization of the effects of infiltration, rather than the result of a physically-based model of rainfall infiltration process. The adopted expression (1) of the factor of safety (and the obtained results, as well as the discussion about them) seem to be deeply affected by this conceptualization.*

The soil hydrological model in TerrSysMP solves 3D Richards equation in the subsurface and computes overland flow in a fully coupled manner with a kinematic wave approximation. The development of a saturated layer is possible at any depth in the soil depending on the soil layering. We will add a description of the hydrological model in section 2.1.3. The second point raised by the reviewer, that wtd might not be the most adequate for the FoS calculation is a fair point. We had tested saturation from TerrSysMP unsuccessfully, but in the process of further exploring to answer this point raised, we solved an issue that was impacting the results. While the results are still not superior to those of the saturation estimated by the conceptual hydrological model PREVAH, they are more informative than the wtd results (e.g. see Figure 1 here below). We will explore the potential of TerrSysMP further and add the results in the manuscript together with a 1:1 comparison with PREVAH's saturation.

[Figure]

*Figure 1. Right, the scatter-plot of different combinations of the rainfall and saturation (from TerrSysMP, as the average saturation in the first two soil layers, 60cm) properties for all landslides (each point corresponds to a landslide): the probability of the saturation in the cell being smaller than the value on the day of the landslide (Pr(Sat<=Sat trig)), the standard deviation in time of the saturation in the cell (std Sat), the ratio between the triggering rainfall intensity (rainfall intensity on the day of the landslide) and the cell mean daily precipitation (Rtrig/mdp), and the difference between the triggering saturation and the temporal mean saturation of the cell (trigg Sat - mean Sat). Left, the histograms of the mean (top) and maximum (bottom) saturation estimated by TerrSysMP during triggering and non-triggering rainfall events, combining spatial (i.e. differences between landslide locations) and temporal (i.e. differences between events in the cells) differences.*

*1. When the soil is considered dry and cohesionless (h=0 in equation (1)), FoS reduces to tan(fi)/tan(beta), which implies that soil depth is ineffective and that everything depends on the quality of your topographic map (beta varies much more than fi, which not surprisingly remains always not far from 30 for the kind of soils you may have in mountain environment).*

*2. When root-cohesion is introduced (by the way, another conceptualization), considering dry slopes with inclination larger than 30, you can easily see that it mostly results FoS>tan(fi)/tan(beta)+0.1c, so that even with the smallest hypothesized cohesion (5 kPa), FoS can be smaller than one only for slopes more inclined than 50.*

*3. When saturated soil cover (h=d) is considered without root cohesion, it is FoS=(ggw)/g\*tan(fi)/tan(beta), that, with the values of g that you assume for the soil (seemingly between 1.2 and 1.6, with gwïA¸Aˇ 1) leads to stability possible only for inclinations smaller than 12, independent of soil depth.*

*4. If we introduce root cohesion when h=d, you get again that for slopes with inclination above 30 some 0.1c is summed up to the previous expression of FoS, that is FoSïA¸Aˇ 0.1c+(g-gw)/g\*tan(fi)/tan(beta), so that only when cohesion is the smallest (c=5 kPa) you may get some slope inclinations for which stability depends on the value of the water table h.*

The referee is describing exactly what we called the unconditionally stable and unstable conditions with the given datasets we used. Perhaps we were not clear in stating that and we will rephrase this part of the manuscript. Our aim was to actually quantify that the range, as the area where hydrology matters and instability is possible, and show that it is restricted to a rather small part of the country when following the FoS approach.

*This given, my overall impression is that all the results from the "physically-based" modeling, namely all the considerations about (un)conditional (un)stable situations, and their comparison with the landslide inventory are strongly affected by the weakness of the model, before than by the issues related to the coarse modeling grid.*

We agree completely with this statement. In fact, the purpose of looking at (un)conditional (un)stable regions (i.e. susceptibility) it's exactly to split the impact of the hydrology (considered in the dynamic comparison instead) and look at the geotechnical model (infinite slope approach) only. The results concerning part (i.e. Figure 3 and Table 1 in the manuscript) are completely independent from the hydrological component and therefore its limitations are only due to the model assumptions and the uncertain input parameters. At the same time, the dynamic soil wetness is affected by the coarse grid resolution, so ultimately it is the combination of the inadequacy of the infinite slope geotechnical model together with the soil wetness state that limits predictions. We will make the separation between hydrological and geotechnical component clearer in the revised manuscript.

*I mention a few points that I believe are worth some discussion: (i) what is the meaning of gamma in equation (1)? This value should change according to soil saturation, and the assumed values between 12 and 16 kN/m3 seem rather to refer to some average field condition (this certainly has an effect on the predicted values of FoS);*

The correct procedure would be to compute it based on the soil bulk density ($\rho_s$), water unit weight ($\gamma_w$), and the void ratio. The spatially variables parameters are $\rho_s$ and the void ratio. While a map is available for the soil bulk density, the void ratio could only be estimated given the soil type. As for the friction angle, the resulting values would be very homogeneous over the country. We therefore simplified by taking the dry soil unit weight value instead. This is a strong simplification, but highlights one of the key messages of this paper, which is that there are strong limitations associated with a physically based approach that should be taken into account when designing a landslide prediction system. We will stress this message and add the explanation of the equation's parameter.

*(ii) to what extent the assumption of the building of a water table is acceptable and consistent with the geomorphological characteristics of the studied alpine slopes (i.e. type of soil and type of bedrock)? (iii) is groundwater table (likely much deeper than the shallow soil covers of interest for the study, as the Authors themselves observe at lines 270-271) an appropriate variable to be chosen for the purpose of this study about shallow landslides? (iv) I guess that TerrSysMP model offers also soil*

*moisture data, so why did you choose groundwater table for your analyses? Concluding on this point, I still believe that the attempt to exploit the information available from a model like TerrSysMP for the sake of predicting landslides is a valuable task, and that it merits to be investigated. But it seems to me that this could be made with more care than it is in this study.*

The answer to this comment can be found above. We will add information about the results considering saturation provided by TerrSysMP also in the revised manuscript (similarly to the plots here above) and a comparison to PREVAH's saturation results.

*On the other hand, there is the conceptual hydrologic model and the use of estimated soil moisture with a probabilistic approach to improve landslide assessment carried out with empirical precipitation thresholds. While this part is more straightforward, there is still a major point that should be clarified. Your aim is to investigate the potential of soil moisture prior the onset of triggering rainfall to improve empirical thresholds. Despite this, from figures 6 and 7 it seems that you never consider this information, as only saturation on the day of the landslide, maximum or mean saturation during an event, and general statistics of the saturation in the cells are calculated. The discussion of the moisture conditions prior the event is limited to graphs of fig. 8, considering mean saturation for 5-60 days long periods preceding rainfall events. Some discussion of the graphs would be worth. For instance: the 5 and 10 days averages seem to be the best choice to correct false alarms (red line well below the others); long events (6 days) seem to lose memory of the effects of initial conditions on missed alarms (all yellow lines drop down for 6 days, while they are above all other lines for shorter event durations).*

The referee is correct in this statement. The point of Figure 6 was indeed to 1) evaluate if the saturation was exceptionally high on landslide days, and 2) see if that related to the triggering rainfall intensity. We considered accounting for saturation over longer periods of time in Figure 6 of the manuscript, but decided not to for two reasons: first, to facilitate comparison to the equivalent Figure 4 of the physically based approach, second, because it would complicate the plot even more. We will consider adding in the revised manuscript or its supplementary material a version of Figure 7, where the saturation over N days prior to the rainfall event is considered. We will also add some more discussion of the different windows prior to landsliding and the effects they may have in rainfall ID prediction in the revised manuscript.

*Instead, in the paper only the brief sentence at lines 303-304 is dedicated to the possibility of building hydrometeorological thresholds, which are just said to be uncapable of improving the performance of precipitation thresholds without any information. I think that much more discussion and data should be presented to the reader, as the effects of prior soil moisture is all in all the focus of the paper.*

What we meant is that replacing duration of the intensity-duration threshold with e.g. saturation (leading to hydrometeorological thresholds) actually lead to worsen performances, while introducing a saturation threshold which splits the event in high and low antecedent saturation followed by two individual ED (cumulative rainfall vs duration) thresholds shows an improvement compared to a unique rainfall threshold. We will add more information about this attempt and the performances obtained.

*In addition to these two major issues, you can find some remarks and comments as annotations in the attached file.*

We report here few comments on the annotations in the pdf:

- We consider indeed predisposing factors as static, as they generally change over very long timeframe. This is very different from the cause (hydrological conditions) and trigger factors which instead we consider to be dynamic. A description of the latter is provided in Bogaard and Greco 2018. We will specify this.
- Fig. 1: a different colormap was suggested to allow better visualization of the friction angle values. Unfortunately, there are really just 2 unique values for the entire country which are visible in yellow and orange.
- We agree that ignoring the increase of soil strength due to soil suction might explain some the results and most importantly the existence of the unconditionally unstable areas. We will add this consideration.

*Fig. 2: Looking at the color maps, it seems that while the slope and elevation dependent depth models result in thinner soil cover in the mountains, the lin. diffusion works in the opposite way around. Also the zoomed pixel apparently shows that the blue-colored (nearly zero depth) lines of the first two models become yellowish in the third model (nearly 2 meters depth). How can you explain this discrepancy?*

Actually the top of the mountains has thin soil cover also in the case of the linear diffusion model (the mountain crests appear like "blue rivers"). Right next to the ridge, the soil often becomes thicker due to the dependence to the second derivative of elevation (curvature).

Bogaard, T. and Greco, R.: Invited perspectives: Hydrological perspectives on precipitation intensity-duration thresholds for landslide initiation: proposing hydro-meteorological thresholds, Nat. Hazards Earth Syst. Sci., 18, 31–39, https://doi.org/10.5194/nhess-18-31-2018, 2018.

---

## Author Response (AR1)

Prof. Carlo De Michele Editor Hydrology and Earth System Sciences 16.06.2021

Dear Prof. De Michele:

Thank you for handling our manuscript entitled *Rainfall-induced shallow landslides and soil wetness: comparison of physically based and probabilistic predictions* (hess-2020-624). We are also grateful to the reviewers for their thoughtful and constructive comments. In response, we have revised the manuscript thoroughly and addressed every comment in the open discussion (point-by-point responses can be found there). In particular, we want to draw your attention to the following general areas of revision:

- 1. We re-did some analyses with new variables, focusing on the hydrometeorological thresholds (Figures 7-9).
- 2. We eased the direct comparison between the two hydrological estimates used (PREVAH and TerrSysMP), adding the same plot considering the saturation provided by each modeling framework (Figure 5).
- 3. We removed parts of the analysis which were confusing the message (e.g., TWI downscaling) and expanded or strengthened others as suggested by the reviewers.

This resulted in new sections of the manuscript, as well as removal of others, and changes in the Figures. All the changes can be found in the "track changes" version of the manuscript submitted. Figures 4 and 6 of the original manuscript were removed, following the reviewers comment about their complexity. Figure 5 was improved (new Figure 4), Figure 7 was simplified and combined with a different plot that allows a direct comparison between the two hydrological estimates (new Figure 5). The new Figures 7, 8, and 9, show the results concerning the exploration of the combination of antecedent wetness and rainfall characteristics for the prediction of landslides in a probabilistic approach, recommended by reviewer 2.

Given these revisions, we are convinced that our revised manuscript is much improved and are pleased to resubmit this version for your further consideration. We hope that you will find it suitable for publication in Hydrology and Earth System Sciences and look forward to hearing your decision.

Thank you and best regards, Elena Leonarduzzi (cc'd Brian W. McArdell and Peter Molnar)

---

## Referee Report (RR1)

General comment: This manuscript presented an interesting topic of comparing two different approaches to predict landslide on a regional scale. Generally, the paper is well presented and comprehensible. There are some minor typing corrections and some further explanations required, as indicated in the manuscript.

Technical comments:

1. Page 12, line 277: how did you obtain this equation and the value of TSS.
2. Page 13, line 280: this conclusion is not always true for six days antecedent saturation before landslides in all plots; i.e., less antecedent saturation governed the miss (T below). Please explain this discrepancy.
3. Page 14, Figure 6: the legend used in all plots should be changed to True positive, miss, false alarm, and true negative.

---

## Author Response (AR2)

**REVIEWER1**

The authors have carefully considered my comments and the issues I raised during the first round of review. I think that this revised version of the manuscript is acceptable for publication in HESS. I invite the authors to consider only the following considerations, that could lead to some minor revision of the Discussion and, maybe, to modify something also in the Conclusions.

The comparison between the landslide predictions carried out with the (coarse-gridded) physically based model, with those obtained with a conceptual hydrological model (with finer resolution), shows that the latter still outperforms the first one, although in this new version the Authors exploit in a better way the outcome of the physically based model and obtain some different indications, compared to the former analyses.

In the Discussion section, they ascribe the limits of the physically based analysis mainly to the coarse resolution (of both the model itself and of input data about geomorphological and geotechnical parameters).

I agree with this intepretation, but I would like the authors to consider also another point (and add some comment in this respect, if they see my point).

They state that the coarse-gridded physically based model cannot reproduce the (lateral) fluxes between cells, as they are too distant from each other (the model grid is 12.5x12.5 km), as the high number of events during which nothing happens to soil moisture seems to demonstrate. Differently, the (conceptual) hydrological model, providing soil moisture at 0.5x0.5 km resolution, gives more valuable information to predict landslides.

What I want to stress here is that in initially unsaturated (shallow) soil covers, the prevailing direction of water fluxes is close to the orthogonal to the ground surface (e.g. Lu et al., 2011), owing to the top (atmosphere) and bottom (whatever it is) boundary conditions, that make the orthogonal hydraulic gradient much much larger than the one parallel to the slope (because all the verticals share more or less the same hydraulic conditions, and so there is very little gradient along the slope (which is at least one order of magnitude longer than the thickness of the soil cover). Only when saturation is reached somewhere within the slope cover, then, in that saturated part, the orthogonal hydraulic gradient becomes of the same order of the one parallel to the slope (depending on slope and bedrock inclination), and so lateral fluxes become significant (leading to subsurface runoff generation). However, I guess that most of the conditionally unstable slopes would have already failed before saturation was reached. I don't expect this picture to change significantly if the model was run over a 0.5x0.5 km gird instead of the 12.5x12.5 grid.

Far from saturation the only way infiltrating water can be drained out of the slope cover is either evapotranspiration (a too slow process over the time scale of 1 to 6 days of rainfall events) or drainage through the soil-bedrock interface, which becomes the most delicate point of the physically based model, about which the authors do not give any information to the reader.

Although I did not look in the cited paper where the PREVAH model is described, I expect that water exchange mechanisms between the three storage modules are somehow introduced in such model, and if the model is somehow calibrated in order to provide reliable results, these mechanisms consider the water exchange between the two upper storage modules and the lower one. This could explain why the dynamics of soil saturation estimated by the PREVAH model better matches with the effects of single rainfall events.

So, at least this is my opinion, the physically based predictions are limited not only by resolution and/or accuracy of input data issues, but also by the unsuitability of the chosen model to correctly assess the effects of one of the (major) processes controlling the dynamics of soil moisture in the unsaturated zone, i.e. the leakage towards the underlying saturated zone.

N. Lu, B.S. Kaya, J.W. Godt (2011). Direction of unsaturated flow in a homogeneous and isotropic hillslope. Water Resources Research 47(2), https://doi.org/10.1029/2010WR010003

We thank the reviewer for the positive comments and the very interesting points raised. We agree that vertical infiltration is dominant in unsaturated soils because the vertical pressure gradients and fluxes are greater than the lateral ones, typically during landslide events (lverson 2000), but we still believe that lateral flow is essential for determining realistic initial conditions for slope failure initiation on hillslopes with some topography (as we showed in Leonarduzzi et el., 2021). In fact, we do not believe that the limiting factor of having the physics-based model run at such coarse resolution is (only) the large distance between cells, but much more the smoothing of topography that destroys possible lateral gradients and of the meteorological forcing. It is hard to conclude this from the experiments carried out here, but we believe that increasing the resolution will indeed significantly improve the representation of water flow, forcing, and consequently the estimates of soil moisture. Furthermore, PREVAH (as most hydrological models) is calibrated for runoff, so it is not explicitly designed for realistically reproducing subsurface storages.

The second point raised, that the physically based model predictions may also be inaccurate because of missing the leakage process in deep soil, is a very good one, and we have added it in the discussion as a possible explanation. Indeed, most models simply assume a homogeneous soil of a certain thickness, or a multilayered soil with prescribed hydraulic conductivity (like we did). However, reality is much more complicated at the interface of the soil and weathered bedrock or saturated layer, where vertical flows can be affected by fractures, preferential flow paths, etc. The conceptual model also does not reproduce exactly these processes, but because it has several soil storage layers, it perhaps manages to get a better soil moisture distribution for this reason. **REVIEWER 2**

This paper studies the prediction of landslide initiation by adding hydrological information to the well-known meteorological thresholds most frequently used and compares probabilistic and physically based modelling approaches for inclusion of antecedent soil wetness state (or water table) in the prediction. This review is on the revised paper. I have not been involved in the first round of reviews.

This paper is timely and contributes to an important research question in landslide research: can landslide forecasting on regional scale be improved by including hydrological information compared to traditional meteorological threshold based forecasting. And if so, how? The paper contains a huge amount of work and data. The methods are sound and clearly described. It studies the obvious question to what extend landslide forecasting can make use of existing regional hydrological model outputs. It tests this using two different regional models with different concepts and resolutions. The underlying work is in my opinion a creative and innovative contribution to this research field. The paper is well structured and written.

Overall, the study concludes that the contribution of the output of an existing large-scale hydrological model with coarse spatial resolution combined with infinite slope FoS calculation in the current state does not lead to an improved landslide forecasting. However, adding the hydrological information from a water balance model (with higher spatial resolution) modestly improve landslide forecasting in Switzerland. It shows that antecedent soil moisture information improves the landslide prediction using the probabilistic framework, but in a quite rudimentary way: by splitting the data sets in dry and wet antecedent conditions (just like proposed frequently before in literature, such as Figure 5.2 published in the Sidle-Ochiai, 2006).

Looking at the first round of reviews, I am of the opinion the authors did a profound job addressing the valuable comments and improved the paper significantly. The fact that this study uses existing models and combines those in landslide forecasting does not mean it is not novel, on the contrary. I think the paper can be accepted for publication in Hess with some minor revisions.

We thank the reviewer for the positive feedback and the insightful comments.

**Comments**

- The authors sometimes use words or abbreviations of words in formula's and in figure caption and axis titles. I personally find those difficult to read and suggest to use 1 symbol for a variable or parameter and subscripts (e.g. x-axis of Fig 4, 5).

Because we are considering both temporal average in the cell and max-min during the specific event, we believe changing this to subscripts will possibly make it simpler but easily to misunderstand. - L256: does a FoS needs to be 'exceptionally' low? I guess just a certain degree lower could be

sufficient? And Figure 4 shows that a bit lower starting point is sufficient.

Yes, it doesn't need to be much lower during triggering events, what we meant to say is that we were trying to find a clear distinction between triggering and non-triggering events, with the FoS being smaller during the former. We removed "exceptionally".

- L278: Would it not be easier for the reader if you use one dominant terminology: False alarm, misses, TP, TN (and later on T, above/below, NT\_above/below). So use TP, TN, FP, FN (and add false alarm or missed alarms in parenthesis to it a few times)? Same for Figure 6.

We agree with this comment and will be consistent in the revised manuscript.

- L298: layout exponent, not in superscript

We have corrected this.

- L300: Figure 8: Honestly, is the TSS really that much different for the different graphs? I would argue you can use all these graphs to split the data set. The entire range of the TSS as function of antecedent wetness is 0.1.... that is not a lot, is it?

This is a commonly raised question. The differences in TSS are generally very small, so a difference of 0.1 is, relatively speaking, not that small. We agree that using a threshold of antecedent moisture over any of the antecedent periods considered would not have made a large difference, but we carried out these experiments exactly to a) check whether this was the case and b) optimize the mean antecedent saturation threshold value.

- L323: vegetation cohesion is not a correct terminology. It is soil cohesion and apparent root cohesion: combined used in the infinite slope model.

We have corrected this.

- L324: for shallow slip surfaces (<2m) cohesion is a very sensitive parameter, by definition. Especially if apparent root cohesion is added.

**We agree.**

- L346: I challenge the authors with this statement. Why would a physically based model per definition (or theoretically) be superior? The so-called physical laws do not apply to the scale of the model but have another Representative Elemental Volume. Equally, could we not argue that on the regional scale we need another set of physical laws. So why can we not say that a physically conceptual model will be theoretically superior to describe the hydrological system on regional scale better than a bottom-up physically based model applied to the 'wrong' scale? At least you should address the scale issue related to the used 'physics'. You could elaborate a bit more on this. That is exactly the point we are trying to make. We are not trying to draw conclusions that go

beyond what we are showing here, but we do say that while the physics-based model is in principle closer to the process representation, at this resolution it is simply incapable of adequately reproducing the soil moisture dynamics that are required for the landslides modeling.

- L360: Maybe the authors can reflect a little why using regional hydrological models have only modest improvement to landslide forecasts and how that compares to hydrometeorological thresholds for landslides using in-situ hydrological (soil moisture) measurements (in Switzerland).

To the authors knowledge nobody has yet combined in situ soil moisture measurements with rainfall observations (hydrometeorological thresholds) in Switzerland, so it is hard to comment on this. The only contribution is that presented in Wicki et al. (2021), where modeled and measured soil moisture is compared for landslide predictions, but soil moisture thresholds are considered, without including rainfall.

- L367: I am a bit surprised to read the first conclusion. The first conclusion is about the effect of cohesion (in your case soil cohesion and apparent root cohesion) or not. I do not consider this the first and most important conclusion. As mentioned above as well, for shallow landslides cohesion can be the dominant resisting force, so putting that to zero is not surprisingly changing the areas where landslides can take place. I suggest to make this not the first conclusion but rather the 3rd or so.

The order of the conclusion matches the order of the research questions in the introduction and the of the results presentation, not the relative importance of them.

- L376: This conclusion is not fully justified, or at least not telling the complete story. Another way of framing this conclusion could be stressing that in your analysis 65% of the country is US (35% when C=0 kpa) That leaves 35% (or 65%) to be potentially 'triggerable'. This is how I read this result.

This of course is true, but the point of this analysis was not really to assess landslide susceptibility, but to see where hydrology can play a role if we choose to follow the infinite slope approach. The idea is that this could help a) understand if the infinite slope approach is suitable at all (by comparing to landslide observations), and b) help constraint the hydrological simulation to a much smaller portion of the country (as there is no need to get saturation estimates in UU or US regions).

**REVIEWER 3**

There are some minor typing corrections and some further explanations required, as indicated in the manuscript.

Technical comments:

1. Page 12, line 277: how did you obtain this equation and the value of TSS.

We briefly explain the procedure in lines 182-184 and reference to our previous work where it is explained more in details. We describe the power law equations as  $E=a^*D^b$  and then test all possible combinations of a and b parameters and pick the one maximizing the TSS.

2. Page 13, line 280: this conclusion is not always true for six days antecedent saturation before landslides in all plots; i.e., less antecedent saturation governed the miss (T below). Please explain this discrepancy.

The reason for this discrepancy is that there are so few triggering events below the threshold of duration 6d that the mean antecedent saturation is statistically not meaningful. In fact, the red dashed line in the lower right panel of Figure 6 drops to basically zero for duration of 6d. We have added a comment to this in the revised manuscript.

3. Page 14, Figure 6: the legend used in all plots should be changed to True positive, miss, false alarm, and true negative.

We have changed this in the revised manuscript.